# Age and Training-Related Changes on Body Composition and Fitness in Male Amateur Cyclists

**DOI:** 10.3390/ijerph19010093

**Published:** 2021-12-23

**Authors:** José Ramón Alvero-Cruz, Jerónimo C. García Romero, Francisco Javier Ordonez, Denis Mongin, Lorena Correas-Gómez, Pantelis T. Nikolaidis, Beat Knechtle

**Affiliations:** 1Sports Medicine and Cycling Training Center, 29004 Malaga, Spain; alvero@uma.es (J.R.A.-C.); jeronimo@uma.es (J.C.G.R.); 2Exercise Physiology Laboratory, Faculty of Medicine, University of Málaga, 29016 Malaga, Spain; 3Faculty of Medicine, University of Cádiz, 11003 Cadiz, Spain; franciscojavier.ordonez@uca.es; 4Quality of Care Unit, University Hospitals of Geneva, 1205 Geneva, Switzerland; Denis.Mongin@unige.ch; 5Faculty of Education Sciences, University of Málaga, 29016 Malaga, Spain; lcg@uma.es; 6School of Health and Caring Sciences, University of West Attica, 122 43 Athens, Greece; pademil@hotmail.com; 7Institute of Primary Care, University of Zurich, 8006 Zurich, Switzerland; 8Medbase St. Gallen Am Vadianplatz, 9000 St. Gallen, Switzerland

**Keywords:** age-related changes, training, fat mass, BMI, cardiorespiratory fitness, cycling, master athletes

## Abstract

Master athletes are considered as a model of healthy aging because they can limit the age-related decline of physiological abilities compared to sedentary individuals. The main objective of this study is to analyze age-related changes and annual training on body composition (BC) and cardiorespiratory fitness (CRF) parameters. The participants in this retrospective cross-sectional study were 176 male cyclists, aged 40–60 years. BC was evaluated through anthropometric measurements and CRF was determined by an incremental cycle ergometer test to exhaustion. A comparative study between age groups was carried out through a one-way ANOVA test and the associations between the variables were assessed by Spearman’s correlation coefficients and multiple regression analysis to estimate the performance. Training was generally associated with a decrease in both body weight and body fat (*p* < 0.05). A decrease in resting heart rate was observed as a vagal effect of kilometers cycled per year (*p* < 0.05). Kilometers cycled per year were associated with an increase in peak power output, which was larger in the master 40 group (*p* < 0.05) with a non-significant upward in VO_2_max (*p* > 0.05). In the performance prediction model, the included variables explained 52% of the variance. In summary, the changes induced by age were minimal in BC and negligible in CRF, whereas HR decreased with age. Training load was generally associated with a decrease in body weight, BMI and body fat percentage that was particularly notable in the abdominal skin folds. A decrease in HRrest was observed as a vagal effect due to kilometers cycled per year, and age did not seem to have a significant effect. The annual cycling kilometers were associated with to high PPO that is greater in the M40 group and a non-significant upward trend in VO_2_max.

## 1. Introduction

Master athletes are considered as a model of successful aging because they are able to maintain high physiological capacities compared to sedentary individuals [1]. Physical exercise and regular training affect maximum oxygen consumption (VO_2_max) and maximum aerobic power (MAP), making these the physiological variables that best correlate with aerobic endurance and peak sports performance, and they are also predictors of cardiovascular mortality [2].

Research has demonstrated that, over the age of 40 years, physiological changes occur with aging, such as alterations in the maximum oxygen uptake and HRmax, and neuromuscular function decreases progressively and significantly [3]. In addition, an age-related decrease in VO_2_max is associated with an increase in mortality [4]. A decline in maximal aerobic power of about 10% per decade in the general population has been described in individuals older than 30 years, although it is not known exactly at what age this occurs. In another study that evaluated the decrease in aerobic power in master athletes between the ages of 37 and 90 years, the decrease was calculated to be between 7 and 14% [5].

Monitoring body composition (BC), and in particular regional adiposity, can identify patterns associated with sports performance and health [6]. Although BC may reflect many factors unrelated to physical activity and training, it is widely recognized that specific low or high adiposity alone can be influential in many different sports and in the performance of competitive athletes. Knowing regional adiposity and BC profiles of athletes can be very useful for coaches, for example, in improving development programs for their athletes and in the longitudinal monitoring of changes in an athlete’s BC, which can indicate sports fitness [7]. Maintaining an ideal BC throughout the year can help to ensure sport performance and can also be a measure of the health and general well-being of athletes.

Similarly, long-term aerobic training produces not only physiological changes in aerobic metabolism, but also general and/or regional modifications in BC and differences can be found according to sex, and this has been reported for female and male triathletes [8]. In addition, improvements in fitness and insulin sensitivity in clinical populations, including those with overweight or obesity, have positive effects on fat distribution [9]. It has been reported that, when comparing continuous aerobic exercise with high-intensity exercise routines after 12 weeks of training, work capacity improved with both types of exercise, with no differences found in abdominal or gluteal circumference or in lipid and biochemical variables [10].

Through continuous and systematic training, master athletes have demonstrated the ability to reduce and limit the declines in physical performance that occur with aging [11,12]. These beneficial effects of continuous participation in competitive sports and systematic training have also been demonstrated in the adaptive capacity of athletes to recover from training stimuli, with most research reporting no significant differences in the recovery kinetics of master athletes and young athletes after exercise [11]. There is growing research interest in master athletes, who are defined as athletes over the age of 35 years who systematically train and compete in organized forms of sport and competition [13].

In the 2015 Spanish national survey of sports habits, cycling was practiced by 38.7% of the Spanish population, of which 47.1% and 28.5% were men and women, respectively [14]. The aim of this work is to determine the changes in BC and cardiorespiratory fitness related to age and training load (kilometers cycled per year) in two groups of amateur cyclists (M40: 40–49.9 years and M50: 50–59.9 years) and to identify the variables that partially explain peak performance. The hypotheses of the present study are that age does not have important effects on BC and that the effect of regular training maintains body fat percentages and cardiorespiratory fitness.

## 2. Materials and Methods

### 2.1. Ethical Approval

The procedures of this study were in accordance with the criteria described in the Declaration of Helsinki [15] and the retrospective study was approved by the ethics committee of the University of Malaga (Report-EMEFYDE-06-2019). The collection and disposal of the data respected the confidentiality of the patients in accordance with the legislation in force. All participants signed an informed consent concerning the procedure.

### 2.2. Participants

In this retrospective cross-sectional study, 176 male amateur cyclists—40–60 years of age, with a median age of 44 (95% CI: 43–45) years, body weight of 73.3 (72.8–74.9) kg, body height of 170.6 (169.0–172.8) cm, BMI of 25.3 (24.8–25.7) kg/m^2^, and a total of 7000 (7000–8000) km cycled per year, in the M40 group (*n* = 123), and a median age of 53 (95% CI: 52–54) years, weight of 75.3 (70.1–78.6) kg, height of 169 (166.9–171.1) cm, BMI of 25.5 (24.8–26.6) kg/m^2^, and a total of 8000 (7000–9965) km cycled per year, in the M50 group (*n* = 53)—participated in the study. The competition level of cyclist was local, and the range of training days per week was 4–6 days.

### 2.3. Experimental Design

A retrospective study was conducted based on laboratory assessments of variables from 2005 to 2015. For the collection of the independent variables associated with the different tests, all the participants underwent a BC evaluation using anthropometric methods and incremental maximal cycle ergometer testing. HRmax was used to verify the degree of maximum effort exerted during the test. To avoid the known errors associated with maximal exercise tests, participants were advised to refrain from strenuous exercise the day before the test and to consume carbohydrates [16,17].

### 2.4. Procedures

#### 2.4.1. Anthropometric Assessment

Within a medical sports study protocol, the following anthropometric data were obtained: body weight was measured on a SECA 813 electronic scale (Hamburg, Germany) to the nearest 0.1 kg, and body height was measured using a wall-mounted SECA 216 stadiometer (Hamburg, Germany) to the nearest 0.1 cm skinfolds (triceps, subscapular, biceps, iliac crest, iliospinale, abdominal, anterior thigh, and medial calf). All skinfolds’ measurements were performed by the same researcher, with a SlimGuide caliper (Rosscraft, Canada) accurate to 1 mm, under the recommendations of the International Society for Advancement of Kinanthropometry [18].

Fat mass was estimated using the Peterson equation [19] with the following equation:% Fat Mass = 20.94878 + (age × 0.1166) − (height × 0.11666) + (sum4sk × 0.42696) − (sum4sk^2^ × 0.00159),(1)
where height is in centimeters and sum4sk is the sum of the triceps, subscapular, suprailiac, and mid-thigh skinfolds.

#### 2.4.2. Cardiorespiratory Fitness Assessment

Cardiorespiratory fitness (CRF) was determined by having the participants perform a continuous incremental exercise test to exhaustion on a mechanical cycle ergometer (Monark ergometer 818E, Monark Exercise AB, Vansbro, Sweden). The exercise test consisted of a warm up of 10 min at 50 W. The test was then started with 25 W/min increases until exhaustion at a cadence of 60 rpm. At completion of the test, each participant performed a 3 min active recovery period in which they continued to pedal at their own cadence with a 50 W load. Heart rate was measured at completion of the 3 min recovery period. All participants were invited and actively encouraged to achieve maximum effort. The environmental parameters were controlled according to International Council of Sport Science and Physical Education guidelines, with a humidity level not exceeding 60% and an ambient temperature of 22–24 °C [20]. The VO_2_max values were calculated using the ratio of HRmax-to-HRrest method and by the equation derived from Uth [21] and validated [22]. Heart rate (at rest, during exercise, and recovery) was determined continuously by an electrocardiographic monitoring system Hellige Cardiotest EK 41 (Freiburg im Breisgau, Germany) and simultaneously by a Polar heart monitor (Polar Electro Oy, Kempele, Oulu, Finland). Finally, peak power output (PPO) was recorded in watts and PPO relative to body weight (PPO in watts/body weight in kg) were calculated. PPO was calculated as the power output corresponding to the highest stage completed, including the fractional representation of any uncompleted stage as followed:PPO = Pcs + ((Tc/60) × 25),(2)
where Pcs is the power output corresponding to the last completed stage and Tc is the time in seconds of the uncompleted stage [23].

#### 2.4.3. Statistical Analyses

Normality was analyzed using the Kolmogorov–Smirnov test. The data are presented as medians and 95% confidence intervals (CI). An association analysis between variables was performed using Spearman’s rank correlation coefficients and the following criteria were adopted to interpret the magnitude of the correlations: r < 0.1, trivial; 0.1 < r ≤ 0.3, small; 0.3 < r ≤ 0.5, moderate; 0.5 < r ≤ 0.7, large; 0.7 < r ≤ 0.9, very large; and r > 0.9, almost perfect. A one-way ANOVA with Student Newman–Keul’s post hoc test or Kruskal–Wallis test with a Conover post hoc test, when appropriate, were applied to test differences between groups. Variables significantly associated with CRF variables (independent variables) were included in a stepwise multiple regression analysis to estimate the predictors of PPO (dependent variables). These variables were selected because they did not violate the collinearity diagnosis (variance inflation factor < 10 and tolerance > 0.2. A statistical analysis was carried out with MedCalc statistical software, version 19.5.3 (Ostend, Belgium). A value of *p* < 0.05 was considered to be statistically significant.

## 3. Results

The BC characteristics and physiological variables of the study groups M40 and M50 can be seen in Table 1. Differences between the groups were observed in age (*p* < 0.05), body height (*p* < 0.05) and fat mass (*p* < 0.0001) with no differences in body weight or BMI (both *p* > 0.05). There were also differences in HRmax and HRrec3 (*p* < 0.001) and in PPO in watts and W/kg (*p* < 0.008 and *p* < 0.05, respectively). No differences were seen in annual training kilometers or maximum oxygen consumption (*p* < 0.05). Percentage changes between groups of −3.3%, −7.3%, and −10.7% were found for HRrest, HRmax, and HRrec3, respectively. Similarly, changes of −7.7%, −6.6%, and −3.6% were observed for PPO, PPO/kg, and VO_2_max, respectively, and, for METs, a change of −3.65% (Table 1).

### 3.1. Correlations between Skin Folds and Age and Training Kilometers

In both the M40 and M50 groups, the correlations between age and the eight skinfolds were low and did not reach significance (*p* > 0.05). Considering the total group, single significant correlations were found only with the triceps, subscapular, and biceps folds (*p* < 0.05). Kilometers cycled per year were inversely and significantly related to most of the skinfolds, especially the trunk folds (*p* < 0.05) (Table 2).

### 3.2. Correlations between Body Composition and Cardiovascular Fitness Variables with Age and km/year

In the study groups, age was not associated with the BC variables, except when considering the total group with fat mass showing a direct correlation (rho = 0.31, *p <* 0.0001), (Table 3, Figure 1). Regarding the CRF variables, low-to-moderate inverse correlations were observed with resting, maximum, and recovery heart rates (all *p* < 0.05 to *p* < 0.0001), (Table 3, Figure 2). Significant correlations were found between PPO/kg and PPO in relation to body weight (*p* < 0.0001). No significant correlation with maximum oxygen consumption was seen, except in the total group (rho = −0.16, *p* < 0.05) (Table 3, Figure 3).

Table 4 shows the multiple regression model that predicts 52% of the variance in the result. In this model, we highlight the variables predicting BC, kilometers cycled per year, HRmax, and VO_2_max (Table 4).

## 4. Discussion

The effects of regular physical training to improve physiological functions and to decrease the deleterious effects that occur with age are well known. This is the first study in a large sample of master cyclists between the ages of 40 and 60 years, in which the effects of age and training volume on BC and CRF variables have been evaluated. In this group of master cyclists, the initial results indicate that, although age in the studied range (40–60 years) does not seem to have a significant effect on the BC and CRF variables, the annual training load has significant and positive effects. The time frame of 20 years from 40 to 60 years is a very long period, but highly trained athletes can nearly maintain their performance during this period of life. A longitudinal analysis of two ultra-triathletes in this period of life showed that the annual decline in the three split disciplines in triathlon were only around 1% per year [24].

The BC and CRF parameters are the most frequently obtained and evaluated for an accurate diagnosis of the level of physical condition and performance in endurance athletes. The cyclists evaluated in the present study would be classified as well-trained athletes, in relation to frequency, duration of training, weekly kilometers, and also from a physiological perspective (PPO and VO_2_max) [25].

The effects of age on body weight show a non-significant downward trend in the M40 group and a slight increase in the M50 group as well as a slight, non-significant upward trend in BMI and fat percentage in both groups. Conversely, the effect of kilometers cycled per year shows a significant downward trend in body weight, BMI, and body fat, which are known effects in athletes of different ages [26].

No studies were found that analyzed the effects of aerobic exercise, and in particular cycling, on the different subcutaneous skinfolds. A systematic review and meta-analysis show that aerobic exercise, strength exercise, and combined exercise produce decreases in subcutaneous and visceral fat tissue [26].

A general increase in trunk fat (largely visceral fat) and a decrease in subcutaneous fat have been observed with aging [27] and are associated with a lower life expectancy [28] and an increase in metabolic diseases as well as an increased risk of cardiovascular disease [29]. In our study, the analysis of the entire sample, from the correlations between the kilometers cycled per year, the highest inverse correlations (rho = −0.31 to −0.39) were observed in the trunk skinfolds, and this would be a positive and desirable effect.

The physiological values of our cyclists were slightly lower than those shown by Peiffer et al., due to their higher training level [23]. Research shows that, above the age of 35–40 years, the main CRF factors, such as VO_2_max and HRmax as well as neuromuscular function [30], begin to decrease, which is calculated to be 10% per decades, but in a progressive manner [31] and similar in individuals despite different levels of physical condition. In our study, the decrease between the M40 and M50 groups was −3.3%, −7.3%, and −10.7% for HRrest, HRmax, and HRrec3 values and values between −7.7%, −6.6%, and −3.6% for PPO, PPO/kg, and VO_2_max, respectively. These data are in accordance with that presented by Peiffer et al. [23]. Regardless of changes in body weight and fat mass, the decrease in VO_2_max may be attributable to a number of factors, such as a loss of oxidative capacity due to muscle changes, in addition to mitochondrial enzymes, and density [1,32]. Other studies showed a smaller decline in performance with age for resistance modalities compared to sprinting, and the largest drop is found for strength events, such as weightlifting and jumping [33].

Associated with the CRF level is the risk of all-cause mortality and morbidity from cardiovascular disease (CVD), which decreased by 2.5% per increase in 1 mL·min^−1^·kg^−1^ with no significant difference between sexes. In this large population study, CVD morbidity and all-cause mortality were inversely related to VO_2_max in all age groups, concluding that increasing cardiorespiratory fitness is a clear public health priority [2].

Aging produces a decrease in HRmax, maximal cardiac output, and maximal arteriovenous O_2_ difference, factors that lead to a decrease in VO_2_max and ultimately reduce sports performance [12]. The decline in VO_2_max with age has been estimated at around 10% per decade from the age of 30 years onwards in healthy sedentary subjects, and, in master athletes, it has been suggested that this can be greater due to higher baseline values [13]. Data from our study show a 3.6% decrease in VO_2_max between the M40 and M50 age groups, values lower than the reported 10% [13].

The drop in endurance performance also seems to depend on the type of locomotion, and this has been proven in certain sports, such as the triathlon, by comparing the three specialties, with a smaller decrease in cycling performance [34]. The great acceptance of the practice of cycling in master athletes can be explained by the benefits of cycling in terms of its efficiency, which diminishes less than in other specialties, such as running.

The importance of a sustained and regular level of training is suggested by studies on cardiovascular fitness and cognition in older adults, which show that a higher level of cardiorespiratory fitness can protect the brain against the effects of aging, mediated by a greater availability of brain oxygenation [35].

CRF reaches its peak between the ages of 20 and 40 years and decreases in both sedentary and trained individuals with increasing age. Recent studies evaluating CRF in large cohorts [36] demonstrated a decrease in all-cause mortality with increasing CRF in a study that included individuals up to an exercise capacity of about 16 METs. In the follow-up study, a high vs. low CRF was associated with a longevity benefit of about 5 years [37]. In the other studies, the reduction in mortality risk resulting from an increase in CRF by 1 MET varied between 10–15%.

Decreases in VO_2_max have a central cause, which is a decrease in maximum cardiac output and HRmax (Fick’s equation) [21]. These two factors are always present, but can vary by age and type of sport. A decrease in HRmax of 0.8 beats·min^−1^·year^−1^ has been described in healthy, active, and highly trained subjects [12]. In our group, the change in HRmax was approximately 1.3 beats·min^−1^·year^−1^. Another cause could be the peripheral factors related to the arteriovenous O_2_ difference, which is associated with a decrease in capillary density and variations in mitochondrial enzymes [1].

The HRrec3 assessment is a measure that expresses the degree of influence of the autonomic nervous system, involving a withdrawal of sympathetic activity and an activation of the parasympathetic system, with the final result being a decrease in HR, thus providing a cardioprotective effect by stabilizing HR values [38]. The effects of age on HRrest did not reach significance in either group, although the decrease in HRmax was greater in the M40 group than in the M50 group. In relation to HRrec, there seemed to be a greater parasympathetic reactivation in the M40 group [38], although parasympathetic reactivation is associated with the degree and intensity of exercise and, therefore, with the contribution of the anaerobic system. The effects of the kilometers cycled per year showed a very similar vagal activation in both groups (identic slopes) on HRrest. HRmax demonstrated a significant downward trend in the M50 group, explained by the effect of age. On HRrec3, a greater parasympathetic reactivation effect was found in the M50 group, which provides a cardioprotective effect [39].

Some studies showed a decline in PPO with age in both untrained, active, and highly trained individuals [40]. In the study by Capelli et al. [41], maximal aerobic power appeared to be maintained in master cyclists until the age of 45 years, with described declines of 1.2% per year [42] or up to 1.5% per year in subjects aged 55–86 years [32]. In our study, maximal aerobic power decreased by 35 W per decade in the M40 group and by 11.6 W per decade in the M50 group, with an intergroup decrease of −7.7%.

Physiological variables obtained in laboratory tests are often associated with endurance cycling and peak power, for example, they are associated with tests that range in distance from 10 to 40 km [43]. These laboratory variables generally include VO_2_max, ventilatory, and lactic thresholds, but, in particular, the predictor variables are the Wingate anaerobic test expressed in W/kg and, most importantly, W/kg at ventilatory threshold 2 [43]. In our study, the anthropometric and BC variables (body height, fat percentage, and BMI) and the CRF variables (VO_2_max and HRmax) predicted PPO and, as a novel element, kilometers cycled per year. Other methods have been described, such as tensiomiography, with the evaluation of neuromuscular variables (contraction times, muscle displacement, and response speed) being moderately capable of predicting PPO [44].

A final consideration is that the evaluation of the different variables in this study was performed using a linear regression model, unlike other works which conducted exponential and/or logarithmic approximations [32]. In our study, the differences with the exponential or logarithmic approximations did not provide a much larger coefficient of determination.

### Strengths and Limitations

One of the main limitations of the study is the indirect calculation of VO_2_max, and, although the equation for the estimation of Uth et al. [21] was internally validated, in our case we do not know the possible bias it could present because there is no direct reference measurement. Another limitation is the cross-sectional nature of the study, which only provided decreasing trends between groups and not a precise change in the CRF and other variables as would a longitudinal study. The anthropometric measurements might be problematic as skinfold testing is not necessarily outdated, but it is highly reliant on the skills of the researcher and then it is put in the equation to obtain an estimate of body composition. Given that this study places a strong emphasis on body composition, it would be of great importance in future studies to use a more sophisticated tool to assess structural characteristics of athletes. A further limitation was that only male subjects were considered. One of the main strengths of the study was that CRF was assessed via maximum effort graded exercise testing, all under the same environmental conditions.

## 5. Conclusions

In summary, in these Masters cyclists of 40–60 years, the effects of age on BC were almost negligible. However, training load was generally associated with a decrease in body weight, BMI, and body fat percentage that was particularly notable in the abdominal skin folds. A decrease in HRrest was observed as a vagal effect due to kilometers cycled per year, and age did not seem to have a significant effect, although it is known to decrease HRmax. Kilometers cycled per year were associated with a high PPO that was greater in the M40 group and a non-significant upward trend in VO_2_max. Age appeared to have a greater effect on PPO in the M40 group and a non-significant downward trend in VO_2_max.

## Figures and Tables

**Figure 1 ijerph-19-00093-f001:**
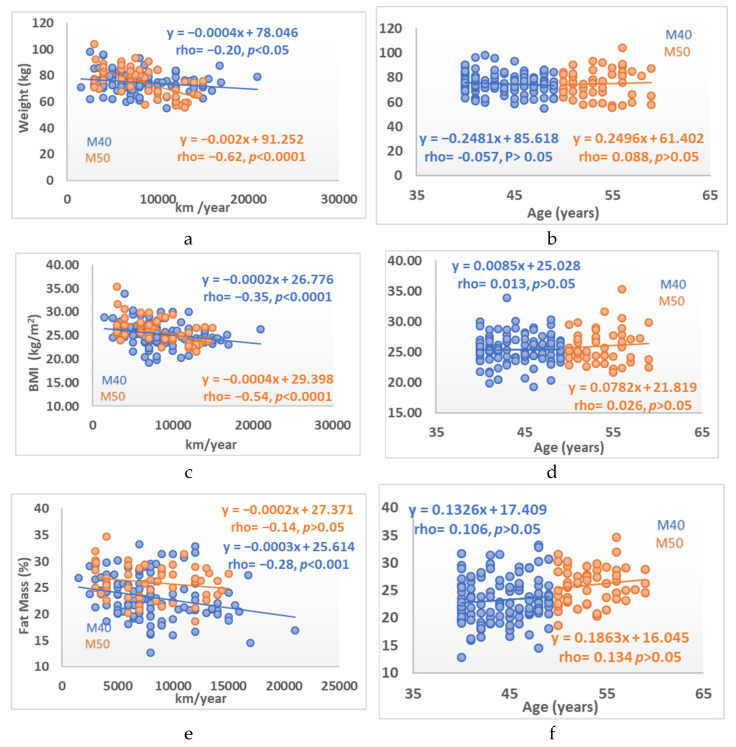
Relationship between body composition (weight, BMI, and fat mass) variables and km/year cycled (**left** column) and age (**right** column). (**a**) body weight versus cycling kilometers, (**b**) body weight versus age, (**c**) BMI versus cycling kilometers, (**d**) BMI versus age, (**e**) fat mass versus cycling kilometers, (**f**) fat mass versus age.

**Figure 2 ijerph-19-00093-f002:**
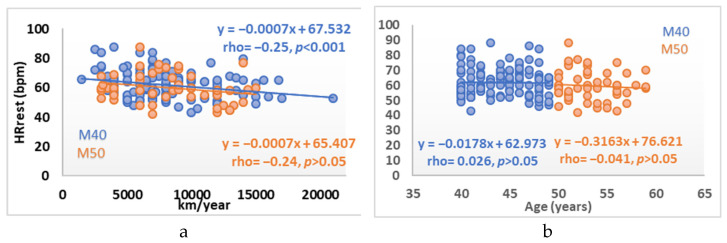
Relationship between heart rate (rest, maximum, and recovery 3 min) variables and km/year cycled (**left** column) and age (**right** column) (**a**) Resting heart rate versus cycling kilometers, (**b**) Resting heart rate versus age, (**c**) Maximum heart rate versus cycling kilometers, (**d**) Maximum heart rate versus age, (**e**) Heart rate recovery 3 min versus cycling kilometers, (**f**) Heart rate recovery 3 min versus age.

**Figure 3 ijerph-19-00093-f003:**
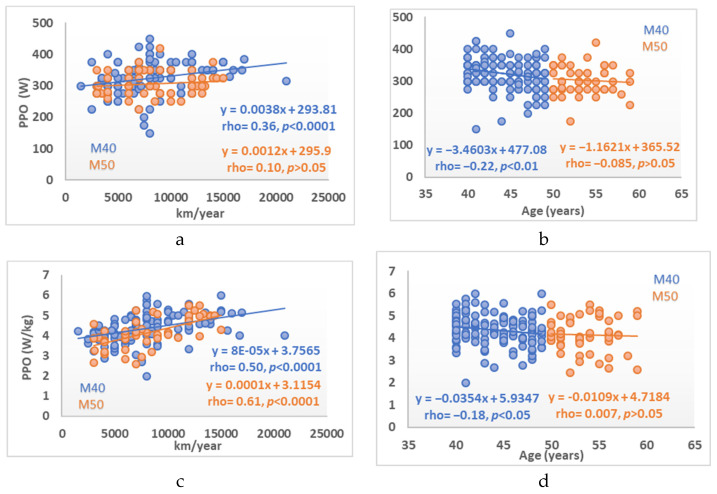
Relationship between cardiorespiratory fitness variables (Peak power output (W and W/kg) and VO_2_max) and km/year cycled (**left** column) and age (**right** column). (**a**) Peak power output versus cycling kilometers, (**b**) Peak power output versus age, (**c**) Peak power output relative to body weigh versus cycling kilometers, (**d**) Peak power output relative to body weight versus age (**e**) Maximal oxygen uptake versus cycling kilometers, (**f**) Maximal oxygen uptake versus age.

**Table 1 ijerph-19-00093-t001:** Body composition and cardiovascular fitness variables of the sample.

Variables	M 40 (*n*= 123)				M 50 (*n*= 53)				*p*	Change (%)
	Min.	Max.	Median	95% CI	Min.	Max.	Median	95% CI		
Age (years)	40	49	44	43.000 to 45.000	50	59	53	52.000 to 54.000	<0.0001	
Weight (kg)	55	98	73.3	72.800 to 74.970	55.4	104	75.3	70.077 to 78.605	0.93	2.73
Height (cm)	156.5	189	170.6	169.000 to 172.800	160.1	184.6	169	166.959 to 171.141	0.039	
BMI (kg/m^2^)	19.28	33.87	25.28	24.778 to 25.722	21.61	35.36	25.56	24.846 to 26.681	0.141	1.11
Fat mass (%)	12.748	33.283	22.695	21.753 to 23.840	18.651	34.696	26.254	24.951 to 27.510	<0.001	15.68
HRrest (bpm)	43	88	61	60.000 to 64.000	42	88	59	56.000 to 60.000	0.114	−3.28
HRmax (bpm)	150	206	178	176.000 to 182.000	142	187	165	162.898 to 170.102	<0.001	−7.30
HRrec3 (bpm)	60	148	112	110.000 to 114.000	60	131	100	96.000 to 108.000	<0.001	−10.71
km/year	1500	21000	7000	7000.000 to 8000.0	3000	15000	8000	7000.0 to 9965.880	0.581	
PPO (W)	150	450	325	325.000 to 350.000	175	420	300	275.000 to 325.000	0.008	−7.69
PPO (W/kg)	2	5.99	4.36	4.237 to 4.567	2.465	5.49	4.072	3.977 to 4.241	0.048	−6.61
VO_2_max (mL/kg/min)	27.614	63.488	44.048	42.403 to 45.192	31.023	60.714	42.458	40.646 to 45.296	0.183	−3.61
METs (mL/kg)	7.89	18.14	12.59	12.118 to 12.916	8.86	17.35	12.13	11.611 to 12.946	0.195	−3.65

BMI: Body mass index; HRrest: Resting heart rate; HRmax: Maximum heart rate; HRrec3: Heart rate recovery 3 min; PPO: Peak power output; VO_2_max: Maximal oxygen uptake; CI: Confidence interval; M40: Masters 40; M50: Masters 50, METs: Metabolic equivalents.

**Table 2 ijerph-19-00093-t002:** Spearman’s rank correlation coefficients between eight skinfolds and age and km/year in all groups.

Group	Variable	Triceps	Subscapular	Biceps	Iliac Crest	Ileospinale	Abdominal	Front Thigh	Medial Calf
All	Age	0.16 *	0.23 **	0.24 **	0.06	0.12	0.11	0.05	0.12
	km/year	−0.23 **	−0.15 *	−0.19 *	−0.32 ***	−0.31 ***	−0.39 ***	−0.13	−0.2 *
M40	Age	−0.004	0.06	0.09	−0.03	0.01	0.05	−0.09	0.01
	km /year	−0.24 *	−0.16	−0.17	−0.32 ***	−0.34 ***	0.38 ***	−0.16	−0.18 *
M50	Age	0.12	−0.07	0.10	−0.02	0.11	0.17	0.11	0.11
	km /year	−0.30 *	−0.24	−0.38 **	−0.35 *	−0.27	−0.48 ***	−0.11	−0.26

M40: Masters 40; M50: Masters 50; * *p* < 0.05, ** *p* < 0.001, *** *p* < 0.0001.

**Table 3 ijerph-19-00093-t003:** Spearman’s rank correlation coefficients between body composition and cardiovascular fitness variables and age and km/year in all groups.

Group	Variable	Weight (kg)	BMI (kg/m^2^)	Fat mass (%)	HRrest (bpm)	HRmax (bpm)	HRrec3 (bpm)	PPO (W)	PPO (W/kg)	VO_2_max (mL·min^−1^·kg^−1^)
All	Age	−0.02	.069	0.31 ***	−0.08	−0.55 ***	−0.34 ***	−0.30 ***	−0.20 **	−0.16 *
	km/year	−0.35 ***	−0.39 ***	−0.20 **	−0.24 **	−0.20 **	−0.36 ***	0.26 **	0.52 ***	0.15
M40	Age	−0.06	0.01	0.11	0.03	−0.29 **	−0.215 **	−0.22 **	−0.18 *	−0.16
	km/year	−0.20 *	−0.35 ***	−0.28 **	−0.25 **	−0.17	−0.26 **	0.36 ***	0.50 ***	0.17
M50	Age	0.09	0.03	0.13	−0.04	−0.38 **	−0.175	−0.085	0.01	−0.08
	km/year	−0.62 ***	−0.54 ***	−0.14	−0.24	−0.32 *	−0.58 ***	0.102	0.61 ***	0.13

M40: Masters 40; M50: Masters 50; BMI: Body mass index; HRrest: Resting heart rate; HRmax: Maximum heart rate; HRrec3: Heart rate recovery 3 min; PPO: Peak power output; VO_2_max: Maximal oxygen uptake; * *p* < 0.05, ** *p* < 0.001, *** *p* < 0.0001.

**Table 4 ijerph-19-00093-t004:** Multiple regression model to predict peak power output (W).

Independent Variables	Coefficient	Std. Error	t	*p*	r_partial_	r_semipartial_	VIF
(Constant)	−4.560.375						
Fat mass (%)	−23.234	0.7978	−2.912	0.0041	−0.2198	0.156	1.627
Height (cm)	31.358	0.4123	7.605	<0.0001	0.5072	0.4075	1.222
km/year	0.003771	0.0007846	4.806	<0.0001	0.3486	0.2575	1.239
HRmax (bpm)	0.706	0.2122	3.327	0.0011	0.2493	0.1783	1.23
BMI (kg/m^2^)	27.619	12.546	2.201	0.0291	0.1679	0.118	1.473
VO_2_max (mL kg^−1^ min^−1^)	16.182	0.427	3.79	0.0002	0.2814	0.2031	1.198

M40: Masters 40; M50: Masters 50; BMI: Body mass index; HRmax: Maximum heart rate; VO_2_max: Maximal oxygen uptake; VIF: Variance inflation factor; R^2^ = 0.52; R^2^ adjusted = 0.503; Multiple correlation coeff. = 0.721; Residual standard deviation = 33.31 W.

## Data Availability

The data presented in this study are available on request from the corresponding author.

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
