# Peer review of "Age and Training-Related Changes on Body Composition and Fitness in Male Amateur Cyclists"

_ijerph, 2021, doi:10.3390/ijerph19010093_

Round 1
Reviewer 1 Report
Standardize the references. There are some as author-year and others as numerical.
line 63-65 - The sentence is not clear.
The study's highlights are evident in the discussion, but I think the main highlights could also be at the end of the introduction.
line 96 - Amateur cyclists are not necessarily cyclotourists.
line 96 to 103 - I wasn't clear why the tracks are so small around average. How were they obtained?
Throughout the text, figures, and tables, km is the correct terminology by the International System, not Km.
line 171 - w/kg -> W/kg
Author Response
Reviewer 1
DEAR REVIEWER
SHORTLY AFTER I SUBMITTED THE MANUSCRIPT, I RECEIVED THE MESSAGE THAT THE FIRST AUTHOR JOSE RAMON ALVERO CRUZ HAD DIED DUE TO A SERIOUS ILLNESS
https://esquelas.diariosur.es/esquela/don-jose-ramon-alvero-cruz-13284.html
ON HIS BEHALF AND ON BEHALF OF THE CO-AUTHORS, I WILL CARRY OUT THE REVISION TO THE BEST OF MY KNOWLEDGE AND BELIEF
Standardize the references. There are some as author-year and others as numerical.
Answer: We agree with the expert reviewer and changes as requested.
line 63-65 - The sentence is not clear.
Answer: We agree with the expert reviewer and changed the sentence to ‘In addition, improvements in fitness and insulin sensitivity in clinical populations, including those with overweight or obesity, have positive effects on fat distribution [9].’.
The study's highlights are evident in the discussion, but I think the main highlights could also be at the end of the introduction.
Answer: We think that the reviewer is thinking of ‘at the end of the abstract’? We adapted the end of the abstract to ‘In summary, the changes induced by age were minimal in BC, negligible in CRF, whereas HR decreased with age. Training load was generally associated with a decrease in body weight, BMI and body fat percentage that was particularly notable in the abdominal skin folds. A decrease in HRrest was observed as a vagal effect due to kilometers cycled per year, and age did not seem to have a significant effect. The annual cycling kilometers were associated with to high PPO that is greater in the M40 group and a non-significant upward trend in VO2max.’
line 96 - Amateur cyclists are not necessarily cyclotourists.
Answer: We agree with the expert reviewer and deleted (also called cyclotourists)
line 96 to 103 - I wasn't clear why the tracks are so small around average. How were they obtained?
Answer: Due to the death of the first author we are not able to add here more details.
Throughout the text, figures, and tables, km is the correct terminology by the International System, not Km.
Answer: We agree with the expert reviewer and changed throughout the manuscript
line 171 - w/kg -> W/kg
Answer: We agree with the expert reviewer and changed as suggested
Reviewer 2 Report
Line 113-125 – This might be problematic as skinfold testing is not necessarily outdated but is highly reliant on the skills of the researcher and then is put in the equation to get an estimate of body composition. Given that this study places strong emphasis on body composition, it would be of great importance to use a more sophisticated tools to assess structural characteristics of athletes. This can be mentioned in the discussion as a major limitation.
Line 222-225 – This is a very interesting finding! How would you explain these results? Twenty-year age difference is significant, but it seems that responsiveness to a training regimen is not age related? Is this the case? Please elaborate a bit more on this and cite other studies that elaborate on similar issues and within similar age group.
Overall, great work!
Congratulations!
Author Response
Reviewer 2
DEAR REVIEWER
SHORTLY AFTER I SUBMITTED THE MANUSCRIPT, I RECEIVED THE MESSAGE THAT THE FIRST AUTHOR JOSE RAMON ALVERO CRUZ HAD DIED DUE TO A SERIOUS ILLNESS
https://esquelas.diariosur.es/esquela/don-jose-ramon-alvero-cruz-13284.html
ON HIS BEHALF AND ON BEHALF OF THE CO-AUTHORS, I WILL CARRY OUT THE REVISION TO THE BEST OF MY KNOWLEDGE AND BELIEF
Line 113-125 – This might be problematic as skinfold testing is not necessarily outdated but is highly reliant on the skills of the researcher and then is put in the equation to get an estimate of body composition. Given that this study places strong emphasis on body composition, it would be of great importance to use a more sophisticated tools to assess structural characteristics of athletes. This can be mentioned in the discussion as a major limitation.
Answer: We agree with the expert reviewer and inserted in the limitations ‘The anthropometric measurements might be problematic as skinfold testing is not necessarily outdated but is highly reliant on the skills of the researcher and then is put in the equation to get an estimate of body composition. Given that this study places strong emphasis on body composition, it would be of great importance in futures studies to use a more sophisticated tool to assess structural characteristics of athletes’
Line 222-225 – This is a very interesting finding! How would you explain these results? Twenty-year age difference is significant, but it seems that responsiveness to a training regimen is not age related? Is this the case? Please elaborate a bit more on this and cite other studies that elaborate on similar issues and within similar age group.
Answer: We agree with the expert reviewer and inserted results of a longitudinal case study of two ultra-triathletes.
Overall, great work!
Congratulations!
Reviewer 3 Report
Thank you so much for inviting me to review this manuscript. Authors tried to to determine the changes in body composition and cardio-respiratory performance related to age and training load (kilometers cycled per year) in two groups of amateur cyclists (M40: 40-49.9 years and M50: 50-59.9 years) and to identify the variables that partially explain peak performance.
Minor comments:
- It would be interesting to include a flow chart, with the process of obtaining the sample.
- Please, consider to replace "cardio-respiratory performance" by "cardiorespiratory fitness (CRF)" throughout the manuscript (e.g., line 81).
- Why were no women included in the study? I think this question should be mentioned in the limitations of the study.
- The title have to include this concern.
- References need to be checked (e.g., Knechtle et al., 2010, Uth et al., 2004).
- Where is reference 8? And 9?
- Moreover, study from reference 8 also assessed females (this must be indicated in the authors' manuscript).
Best regards,
Author Response
Reviewer 3
DEAR REVIEWER
SHORTLY AFTER I SUBMITTED THE MANUSCRIPT, I RECEIVED THE MESSAGE THAT THE FIRST AUTHOR JOSE RAMON ALVERO CRUZ HAD DIED DUE TO A SERIOUS ILLNESS
https://esquelas.diariosur.es/esquela/don-jose-ramon-alvero-cruz-13284.html
ON HIS BEHALF AND ON BEHALF OF THE CO-AUTHORS, I WILL CARRY OUT THE REVISION TO THE BEST OF MY KNOWLEDGE AND BELIEF
Thank you so much for inviting me to review this manuscript. Authors tried to determine the changes in body composition and cardio-respiratory performance related to age and training load (kilometers cycled per year) in two groups of amateur cyclists (M40: 40-49.9 years and M50: 50-59.9 years) and to identify the variables that partially explain peak performance.
Minor comments:
It would be interesting to include a flow chart, with the process of obtaining the sample.
Answer: We thank the expert reviewer for this suggestion. Due to the early death of the first author, we have no possibility to get these details.
Please, consider to replace "cardio-respiratory performance" by "cardiorespiratory fitness (CRF)" throughout the manuscript (e.g., line 81).
Answer: We agree with the expert reviewer and changed as suggested.
Why were no women included in the study? I think this question should be mentioned in the limitations of the study.
Answer: We agree with the expert reviewer and add this aspect in the limitations.
The title has to include this concern.
Answer: We agree with the expert reviewer and change the title to’ Age and Training-Related Changes on Body Composition and Fitness in Male Amateur Cyclists’
References need to be checked (e.g., Knechtle et al., 2010, Uth et al., 2004).
Answer: We agree with the expert reviewer and checked all references.
Where is reference 8? And 9?
Answer: We agree with the expert reviewer and checked all references.
Moreover, study from reference 8 also assessed females (this must be indicated in the authors' manuscript).
Answer: We agree with the expert reviewer and mention this aspect in the text
Best regards,